# Bacteriophage–Host Interactions and the Therapeutic Potential of Bacteriophages

**DOI:** 10.3390/v16030478

**Published:** 2024-03-20

**Authors:** Leon M. T. Dicks, Wian Vermeulen

**Affiliations:** Department of Microbiology, Stellenbosch University, Stellenbosch 7600, South Africa; 23130709@sun.ac.za

**Keywords:** bacteriophage-derived proteins, therapeutic potential

## Abstract

Healthcare faces a major problem with the increased emergence of antimicrobial resistance due to over-prescribing antibiotics. Bacteriophages may provide a solution to the treatment of bacterial infections given their specificity. Enzymes such as endolysins, exolysins, endopeptidases, endosialidases, and depolymerases produced by phages interact with bacterial surfaces, cell wall components, and exopolysaccharides, and may even destroy biofilms. Enzymatic cleavage of the host cell envelope components exposes specific receptors required for phage adhesion. Gram-positive bacteria are susceptible to phage infiltration through their peptidoglycan, cell wall teichoic acid (WTA), lipoteichoic acids (LTAs), and flagella. In Gram-negative bacteria, lipopolysaccharides (LPSs), pili, and capsules serve as targets. Defense mechanisms used by bacteria differ and include physical barriers (e.g., capsules) or endogenous mechanisms such as clustered regularly interspaced palindromic repeat (CRISPR)-associated protein (Cas) systems. Phage proteins stimulate immune responses against specific pathogens and improve antibiotic susceptibility. This review discusses the attachment of phages to bacterial cells, the penetration of bacterial cells, the use of phages in the treatment of bacterial infections, and the limitations of phage therapy. The therapeutic potential of phage-derived proteins and the impact that genomically engineered phages may have in the treatment of infections are summarized.

## 1. Introduction

Per annum, approximately 700,000 deaths related to multidrug-resistant infections are reported worldwide, which may increase to 10 million by 2050 [1]. Treatment with bacteriophages (phages) differs from treatment with antibiotics in that phages are mostly strain-specific, but some phages can infect several strains within a species, and others can infect strains (or members) from multiple genera [2,3]. Phage therapy may be used as an alternative to treatment with antibiotics. Bacteria can, however, develop resistance to phage attacks. This is achieved through the modification (or loss) of phage receptors or the secretion of molecules that block adhesion sites [4,5]. *Bordetella bronchiseptica* uses reverse transcriptase to protect cells against phage infection [6].

The biggest advantage of using phages to treat infections lies in their ability to produce enzymes that interact with bacterial surfaces or destroy biofilms. These enzymes also play an important role in the entry and exit of phages from susceptible hosts [7,8,9]. Endogenous lysins are produced during the final phase of a bacteriophage infecting a bacterial cell. The proteins holin and spanin, encoded by genes in the lysin operon, assist with the translocation of lysin across the cell membrane to the peptidoglycan layer. Cell lysis is initiated when the holin forms micron-scale holes in the inner membrane, releasing active endolysin into the periplasm to degrade the peptidoglycan. Spanin is involved in the outer membrane disruption of Gram-negative bacteria and forms a protein bridge that connects both membranes [10]. Lysins are grouped into six main categories, i.e., N-acetyl-β-D-glucosaminidase, N-acetyl-β-D-muramidase (lysozyme), lytic transglycosylase, N-acetylmuramoyl-L-alanine amidase, L-alanoyl-D-glutamate endopeptidase, and D-alanyl-glycyl endopeptidase [11]. Endopeptidases degrade protein moieties and amidases (e.g., N-acetylmuramoyl-L-alanine amidase) hydrolyze amide bonds between glycans and peptides [7]. Exolysins, classified as hydrolases or lyases, on phage tail fibers, tail spike proteins (TSPs), and phage baseplates cleave exopolysaccharides (EPSs) on the surface of bacterial cells to facilitate adsorption and degrade capsule polysaccharides (CPSs) [7]. Some phages have endosialidases associated with tail structures to degrade polysaccharide barriers. Phage polysaccharide lyases cleave 1,4 glycosidic bonds in hyaluronate, alginate, and pectin [7]. Lipases, produced by a select few phages, hydrolyze the carboxyl ester bonds of triacylglycerols, resulting in the release of organic acids and glycerol [12,13]. Depolymerases of phages destroy biofilms to gain access to bacterial cells [14].

This review summarizes the interactions between phages and bacteria, and the resistance mechanisms that bacteria have developed against phage attacks, including immune systems. The role that phages and phage-derived proteins play in the fight against bacterial infections is discussed, and the limitations of phage therapy are highlighted. The improvement of phage infections using genetic engineering is investigated.

## 2. Classification of Bacteriophages

Phages were originally classified into four basic morphological groups, i.e., tailed (order *Caudovirales*), polyhedral (*Microviridae*), filamentous (*Inoviridae*), and pleomorphic (*Plasmaviridae*) [15,16]. For more information on the older classification system of phages, the reader is referred to the ninth report of the International Committee on Taxonomy of Viruses (ICTV), available at https://ictv.global/report_9th (accessed on 5 February 2024). In a recent proposal by the International Committee on Taxonomy of Viruses (ICTV), the families *Podoviridae*, *Siphoviridae*, and *Myoviridae* as well as the order *Caudovirales* were abolished, and a binomial system of nomenclature for species was proposed that included gene transfer agents (GTAs) in the taxonomic framework by classifying them as viriforms. This led to the creation of one class, seven orders, 31 families, 214 genera, and 858 species [17]. Due to the recently mandated binomial nomenclature format, 8982 of the current 11,273 species now have binomial names. For further information, the reader is referred to Zerbini et al. [17]. Phage genomes are either single-stranded (ss)DNA, double-stranded (ds)DNA, dsRNA, or ssRNA. Genome sizes vary, ranging from 3.3 kbp in *Leviviridae* to 500 kbp in *Bacillus megaterium* phage G [18]. In some cases, archaeal viruses and phages share a similar morphology, as observed in *Inoviridae* [19]. Archaeal viruses and phages likely evolved from a common ancestor that infected similar hosts before the divergence of bacteria, archaea, and eukarya [20].

The specificity of phages differs; for example, *Rhizobium etli* phage ph09 has a narrow host range and infects only 4 strains within the species, contrary to *Staphylococcus aureus* phage ϕ812 that infects 743 strains, including 38 coagulase-negative *Staphylococcus* spp. [21,22]. Examples of phages infecting hosts from different genera have also been described, e.g., the promiscuous podophage Atoyac that infects species of the genera *Aeromonas*, *Pseudomonas*, *Yersinia*, *Hafnia*, *Escherichia*, and *Serratia* [23].

## 3. Phage–Host Adsorption and Cell Entry Strategies

Phages are obligate parasites and are reliant on bacteria to complete their lifecycle. Bacteria have developed strategies to negate phage infections. Host defense strategies can be categorized into exogenous, physical barriers, and endogenous defensive mechanisms, e.g., clustered regularly interspaced palindromic repeat (CRISPR)-associated protein (Cas) systems, restriction modifications, and abortive infections. These strategies are often overcome by phages through mutation or changes by hosts to block certain phages, making them susceptible to other phages targeting different or modified receptors. In addition, phages can modulate host virulence, allowing for the survival of their hosts and subsequently, the phages when their hosts are under threat from the human immune system [24]. Phages are known to inactivate host-specific RNA polymerases that inhibit host translation, and some engage in superinfection exclusion, whereby they inhibit further infection of their host by other phages, including the same phages [25,26]. Some modulate receptors are used by other phages, thus blocking entry and preventing secondary infection of their host. The co-evolution of bacteria to resist phage attacks can impact host growth, virulence, and environmental fitness [27].

The adsorption of phages to the host involves a series of interactions between binding proteins of the phage and receptors on the surface of the host cell [28]. An example of a tailed phage with primary receptors on long tail fibers attaching to bacterial surfaces is shown in Figure 1. Attachment to a host may be due to Brownian motion, dispersion, or flow [29]. Reversible binding allows for the desorption of a phage particle and infection of another cell from the same strain in the culture. Once suitable binding of primary receptors is achieved, irreversible binding of the phage injection machinery is initiated. Enzymatic cleavage of the host cell envelope components reveals specific receptors required by the phage. This leads to conformational changes in phage tail machinery proteins, allowing for the ejection of the phage genome into the host cell [28]. For some phages, e.g., T5, primary adsorption involves the O-antigen polymannose moiety of lipopolysaccharides (LPSs). Irreversible binding occurs at the conical portions of the straight tail which harbor receptor-binding proteins that attach permanently to the ferrichrome outer membrane protein, FhuA [30]. Outer membrane proteins are often hijacked by phages as target sites for receptor-binding proteins (RBPs).

Attachment to Gram-positive bacteria differs from the attachment to Gram-negative bacteria because of variations in the thickness of peptidoglycan layers and the levels of lipoteichoic acids (LTAs), lipopolysaccharides (LPSs), and lipoproteins [31]. The different cellular structures facilitating the attachment of phages to bacterial cells are shown in Figure 2. Phage SSU5, isolated from O-antigen-deficient *Salmonella* mutants, could not infect wild-type (WT) *Salmonella* due to the masking of core polysaccharides by the O-antigen [32]. In contrast, O-antigen-specific phages could not infect O-antigen mutants lacking receptors for core polysaccharides. Specificity is key to phage infections and is particularly apparent for Mu G(+), which targets the terminal Glcα-2Glcα1 or GlcNAcα1-2Glcα1 within LPSs [33]. Saccharide moieties are abundant in some Gram-positive bacteria and many phages have taken advantage of these, making them suitable sites for host attachment.

A siphovirus LL-H phage targets glucose moieties of LTA and irreversibly binds to a glycerol phosphate group in the LTA of *Lactobacillus delbrueckii* [34]. Glucose, rhamnose, and galactose in cell wall pneumococcal capsular polysaccharides (PPSs) have been used by phages as receptor sites [35]. Phages can also bind directly to peptidoglycan, as observed for *Listeria* phage A511, which exhibits a broad host range due to the conserved nature of peptidoglycan [36]. In many cases, phages need to gain access to the underlying structure of the bacterial cell wall and degrade a path through the cell envelope to access the necessary receptors responsible for viral entry. This is performed by using LPS-specific glycanases and deacetylases. Similarly, phages that infect Gram-positive bacteria produce exolysins that degrade peptidoglycan layers [37,38]. In Gram-positive bacteria, teichoic acid in the cell wall (WTA) provides strength, hydrophobicity, and zwitterionic properties to attract cations such as Ca^++^ and K^+^ and may also serve as adhesins to attach to surfaces and other bacteria. A SPP1 phage infecting *Bacillus subtilis* targets WTA as its primary reversible receptor and then facilitates irreversible adsorption to its host by a secretory system protein YueB [39]. Other mechanisms that play a role in infection are the transmission of signals, and tactics used to inject the virus genome [40].

Some phages target variable cellular surface structures such as pili, flagella, and polysaccharides. Other phages are plasmid-dependent and will only infect cells carrying, and expressing, genes involved in phage entry [41,42]. Flagellotropic phages with chi-like tail fibers target H-antigens on flagella, as observed for *Salmonella* serovars [43,44]. *Klebsiella pneumoniae* is well known for its ability to produce exopolysaccharides. Phages attacking *Klebsiella* produce depolymerases that degrade the polysaccharide layer. An example is depoKP36, which degrades capsules and exposes underlying structures required for phage infection [45]. Modification of capsules, or loss of capsule formation, results in the development of phage resistance, as demonstrated by Song et al. [46]. Several phages of *Leviviridae*, *Inoviridae,* and *Cystoviridae* bind to conjugative pili, Type IV pili, and other pili involved in bacterial attachment [47,48,49]. Jalasvuori et al. [50] investigated plasmid-dependent bacteriophages to initiate antibiotic susceptibility in previously resistant strains of *Salmonella enterica* and *E. coli*. The plasmid-dependent phage PRD1 drives phage resistance by causing a loss in plasmids containing antibiotic-resistant genes. This allows bacteria to avoid phage infection by silencing the expression of phage receptors.

Once attachment to the host and degradation of the host’s surrounding cell barriers are achieved, the genetic material of the phage enters the host’s cytoplasm. Tailed phages use specialized tail structures to deliver their genome into the host with variations occurring across the three subfamilies. Myoviruses that infect Gram-negative bacteria allow for conformational changes to their baseplate and trigger the expulsion of a rigid internal tube, acting as a channel crossing the host cell’s envelope. This is followed by fusion with the cell membrane, driving the phage genome from the capsid into the host’s cytoplasm [51]. Several phages that infect Gram-positive (and Gram-negative) bacteria use specialized tape measure proteins and the opening of a proximal plug that joins the capsid with the tail and then releases the phage genome [52]. Some tail fiber structures produce murein hydrolase that degrades peptidoglycan [53]. Phage T5, a siphovirus phage, uses tails to interact with the iron-siderophore receptor FhuA on the host, resulting in conformational changes. This leads to localized degradation of peptidoglycan and pore formation, causing fusion of the host’s outer and inner membranes and providing safety to viral DNA from periplasmic nucleases. It also creates a pore through which the genetic material is released into the host’s cytoplasm [54]. In essence, much is the same for podoviruses that infect Gram-negative hosts. Once their tail tip reaches the cell membrane, it forms a conduit for genome injection. In Gram-positive hosts, it is believed that podoviruses burrow a tunnel into the host’s thick peptidoglycan layer with exolysins reaching the cell membrane surface [55].

The filamentous M13 phage enters the host by binding to the host pilus. Then, the pilus retracts toward the host’s inner membrane, bringing the phage along with it [56]. The phage’s two N-terminal domains are involved in binding to primary and secondary receptors on the host, whilst the C-terminal domain is responsible for virion uncoating and facilitating the release of DNA into the host’s cytoplasm. Tectivirus PRD1 is an unusual phage in that, instead of a tail, it uses the internal membrane acquired from the host during virus assembly to inject its DNA [57]. The phage’s capsid protein, P2, is used for receptor recognition and initiates irreversible binding. A P11 protein is responsible for outer membrane penetration, and a P7 protein digests the peptidoglycan layers after which it is unclear whether cell membrane penetration or fusion occurs. Finally, there is evidence that a tubular structure forms between the capsid and host cell membrane, which facilitates DNA injection [58]. Members of the *Cystoviridae* have a lipid-rich envelope encapsulating their nucleocapsid and are believed to have a similar mechanism of entry to animal viruses. Adsorption to the host is mediated through pilus retraction. The phage envelope and host outer membrane fuse together, followed by peptidoglycan degradation allowing the nucleocapsid to enter the periplasmic space. An invagination of the host cell membrane occurs, facilitating entry of the virion into the host cytoplasm in an endocytic-like vesicle. Internal core particles containing the phage genome are released into the host cytoplasm, and the segmented dsRNA is polycistronically transcribed into mRNA by viral RNA-dependent RNA polymerase [59,60]. All of this is performed within the core particle and exported out, avoiding host antiviral mechanisms, and providing a template for the translation of viral proteins.

## 4. Resistance to Phage Attack

As bacteria evolve, they generate mechanisms to avoid phage infection but in doing so they encourage many phages to evolve counter mechanisms. Hosts may change their cell walls such that they express new RBPs to combat phage adsorption. In this instance, phages can adapt to these changes. For example, an λ phage typically binds to LamB but *E. coli* can diverge to express a new receptor, OmpF. However, the λ phage can evolve to sustain its cell tropism. This is achieved through a mutation to the distal domain of the J protein, which in turn is achieved through a combination of mutations to the RBP gene [61]. Some bacteria that produce capsules or exopolysaccharide structures hinder the accessibility to RBPs necessary for phage infection. Interestingly, a strain of capsulated *E. coli* EV36 was found to avoid T7 phage infection using its K1 capsule, and subsequent removal of the capsule led to plaque formation [62]. Some phages can produce a depolymerizing enzyme with the ability to degrade the host capsule. This has been observed in coliphages, K1F and K1-5, which possess endosialidases that hydrolyze K1 capsules [62]. The endosialidase, encoded by *gp143*, appears at the distal portion of the tail and is one of five spike proteins that facilitates accessibility for other RBPs to interact with host receptors [63]. Bacterial capsules differ vastly. *Streptococcus pyogenes* synthesizes a capsule composed of hyaluronic acid, whilst *Pseudomonas* spp. produce exopolysaccharide capsules. *Klebsiella* spp. produce 77 distinct capsule serotypes which differ in their capsular monomers, stereochemistry, and glycosidic linkages [64]. These bacteria are still susceptible to phages, but they require different depolymerases. Examples are *Streptococcus* phage H4489A, which produces hyaluronidase, and *Pseudomonas* phage AF, which contains an exopolysaccharide hydrolase [65,66]. Alternatively, bacteria can avoid infection by expressing their phage target surface proteins in a stochastic manner during phase variations or physiological regulations, but phages can counter this by relying on alternative receptors or target host surface structures that are a necessity to the host [67]. An example of the latter was observed in *Campylobacter jejuni,* which avoided phage infection by phase-variable expression of the O-methyl phosphoramidate (MeOPN) moiety in its capsule [68]. Furthermore, Sørensen et al. [68] observed acquired resistance in vivo as phage-resistant *C*. *jejuni* were selected that either lacked the MeOPN or had gained a 6-O-Me group on the capsule. Lastly, some *Vibrio* and *Escherichia* spp. deploy extracellular vesicle decoys displaying RBPs on their surfaces that adsorb phages from the environment, lowering the environmental phage titer and lowering host exposure [69].

Although phage therapy is seen as a solution to the treatment of infections caused by antibiotic-resistant bacteria, several challenges remain, such as selecting the most adequate phage(s) against a given infection, the risk of phage resistance development, and the immune response to phages by the host, as well as novel regulatory requirements. Although we have a better understanding of the mechanisms behind phage resistance, many defense systems remain uncharacterized or yet undiscovered. Much more research is required to have a complete understanding of how bacteria develop resistance to phages, especially in the context of the human body.

## 5. Bacterial Immunity to Phage Infections

Immune systems are not only present in complex multicellular organisms. Prokaryotes also have a primitive immune system conferring adaptive immunity against bacteriophages. Adaptive immune systems such as CRISPR-Cas target and degrade nucleic acids derived from bacteriophages and other foreign genetic elements, whereas innate immune systems rely more on restriction modifications, DNA degradation systems, and abortive infection [70]. More than 80% of bacterial genomes respond to incoming viral infections with at least one restriction–modification system [71]. Using modification enzymes, bacteria can methylate their own DNA, protecting it from restriction endonuclease activity, which cleaves any unmethylated DNA such as phage DNA. To overcome this defense mechanism, phages can encode their own strain-specific modifying enzymes that methylate specific sequences of the phage DNA, thereby preventing degradation by bacterial endonucleases. Many T1 phages mask the recognition sites required by the restriction enzymes *Sau3AI* and *DpnI* by methylation using Dam methyltransferase during genome packaging [72]. The T7 phage inhibits restriction enzyme activity with an anti-restriction protein, Ocr (overcome classical restriction), an early-expressed protein that blocks the active site of type I DNA restriction enzymes by structurally mimicking the phosphate backbone of bent B-form DNA [73].

Alternatively, some *E. coli* and *Salmonella* spp. can express nucleases that actively seek out DNA with free ends. The RecBCD enzyme is a helicase–nuclease responsible for dsDNA repair but also protects bacteria by degrading invading linear DNA from phages and extra-chromosomal DNA [74]. This can be countered using gp2, found in T4 phages, which attaches to the free-ended phage DNA, thereby blocking accessibility of RecBCD’s active site. Similar activity has been observed in Lambda and Mu phages by the Gam protein [75,76]. The CRISPR/Cas-associated genes have been acknowledged as the DNA-encoded and RNA-mediated adaptive immune system of bacteria [77]. This process occurs in three stages, namely adaption, CRISPR RNA (crRNA) biosynthesis, and targeting [78]. During adaptation, invading phage nucleic acids are integrated into a CRISPR array, comprising CRISPR loci (21–48 bp) direct repeats interspaced by the newly acquired phage nucleic acids known as CRISPR spacers. This array is then translated and processed by Cas endoribonucleases within the repeated sequences to synthesize small crRNAs. It is important to note that the crRNA and Cas proteins form a complex complementary to incoming phages, which in turn are identical to the parent phage. This phenomenon induces a sequence-specific cleavage of phage nucleic acids, preventing the proliferation of the phage genome and viral progeny [70,77]. Evading adaptive immunity is not only seen in eukaryotes but has also been reported in bacteriophages. It has been observed that *Streptococcus thermophilus* phages overcome their host’s resistance in the form of a newly gained spacer. This is facilitated by a single point mutation in the target spacer that inhibits the formation of Cas protein–crRNA complexes, impacting the specificity of the Cas protein and allowing the phage target to continue proliferating within the host [78]. A *Vibrio cholera* phage, ICP1, counteracts the host antiviral defense islands, phage-inducible chromosomal island (PICI)-like elements (PLEs), by encoding a CRISPR/Cas-like system of its own to mediate DNA cleavage and destruction of PLEs [79]. Many more anti-CRISPR genes involved in host defense evasion have been extensively investigated by Pawluk et al. [80], who showed that some *Pseudomonas* phages carry multiple mechanisms to combat different types of CRISPR systems [80,81].

Abortive infection mechanisms instigated by hosts are diverse. In *Lactococcus lactis*, a gene related to abortive infection (*abiZ*) is known to accelerate the lysis process within the host before matured viral progeny can be produced [82]. Other mechanisms causing abortive infection are *E. coli* Lit proteins inhibiting translation, and Rex proteins A and B recognizing the phage DNA–protein complex, thus facilitating membrane depolarization and ultimately cell death [83,84]. These mechanisms are deployed by ‘sacrificial’ bacteria to prevent further infection of the entire population.

## 6. Bacteriophage-Based Therapeutics

### 6.1. Phage Therapy

Examples of phage therapy and phage-associated clinical trials with an impact on human health are listed in Table 1. Studies in humans are mostly limited to case reports. Only a few randomized, placebo-controlled trials have been reported. Whilst a few trials have shown that phages are safe therapeutic agents, they often do not supersede the standard-of-care (SOC) antibiotics or conventional treatments used in disease management (Table 1). In a clinical trial using phages to treat complicated urinary tract infections (UTIs), the placebo and treatment with antibiotics resulted in a 37% and 28% success rate, respectively. This superseded the 18% success rate reported with phage treatment [85]. In-depth and well-designed clinical trials are required to assess the efficacy of phage therapy and phage–antibiotic therapy.

Phages were successfully used in the treatment of a multidrug-resistant *Acinetobacter baumanii* infection [86] and the eradication of *P. aeruginosa* from aortic grafts [87]. In the latter study, the patient was treated with a lytic phage (OMKO1) bound to the outer membrane protein Mof and the mexAB and mexXY multidrug efflux systems of *P. aeruginosa* [88]. Targeting these efflux pumps increased the susceptibility of the pathogen to ceftazidime by two-fold and to ciprofloxacin by ten-fold. In addition to the impact of OMKO1 on antibiotic susceptibility, the phage destroyed the biofilms deposited on the implanted device [88]. A patient suffering from cystic fibrosis (CF) who underwent a bilateral orthotopic lung transplant to eradicate a chronic *Mycobacterium abscessus* pulmonary infection did not respond to treatment with antimycobacterial agents [89]. Treatment with a phage cocktail, also active against *Mycobacterium smegmatis,* eradicated *M. abscessus*. The repressor gene from two of the most strain-specific phages in the cocktail had to be deleted to convert the phages from temperate to lytic. For a listing of clinical data published from 2000 to 2021 that involved 2,241 patients who were treated with phage therapy, and the safety and efficacy of phage therapy, the reader is referred to the review by Uyttebroek et al. [90]. In this review, the authors summarize the effect of phage therapy in pneumology, urology, orthopedics, dermatology, otorhinolaryngology, ophthalmology, gastroenterology, cardiology, and intensive care medicine. Clinical improvement was seen in 79% of patients and bacterial eradication in 87% of patients who were on phage therapy. Case studies of phage therapy and the outcome of the results are listed in Table 1 and Table 2.

Bacteria that develop extreme resistance to phage treatment may be eradicated by using live lytic, bioengineered, phage-encoded biological products in combination with antibiotics to treat bacterial infections [91]. These phages must be screened for the absence of resistant genes, virulent genes, cytotoxicity, and interaction with host tissue. Temperate phages are usually not used, as they may transfer antibiotic resistance genes to bacteria and lead to the development of genetically altered, or extremely resistant, pathogens [92,93,94]. Genes encoding resistance to β-lactams (blaTEM), fluoroquinolones (qnrS), macrolides (ermB), sulphonamides (sulI), and tetracyclines (tetW) have been detected in the genomes of phages present in activated sludge [95], urban wastewater, hospitals [96,97,98,99], freshwater fish [96], and human feces [100].

Reports of fluoroquinolones and anticoagulants that induce the expression of prophage genes and the spreading of temperate phages [101] are alarming and may influence phage therapy in the future. Colomer-Lluchn et al. [102] have shown that treatment of wastewater with ethylenediaminetetraacetic acid (EDTA) or sodium citrate activates the lytic cycle of lysogenic phages, which increases the release of phages from infected cells and the spreading of antibiotic resistance genes located on virus genomes. Phages isolated from patients infected with antibiotic-resistant bacteria may carry genes encoding resistance to the same antibiotics. This was observed in phages isolated from patients with CF who received extensive antibiotic treatment [103]. The authors identified 66 genes that may each encode an antibiotic efflux pump. Of these, fifteen genes encoded resistance to fluoroquinolone and nine to β-lactam antibiotics. Although these findings are of major concern, other studies have shown that the risk of transduction, although possible, is lower than originally anticipated. Enault et al. [104] argued that genes encoding antibiotic resistance are not commonly found in the genome of viruses [104]. Furthermore, the methods that are used to detect antibiotic resistance genes in virus genomes have been questioned [105].viruses-16-00478-t001_Table 1Table 1Past and current phage therapy and phage-associated clinical trials with an impact on human health (updated from Abedon et al. [106]).Infection(s)/Phage Trial InterestCausative Agent(s)/Agents of InterestOutcomes/CommentsReference/Clinical Trial IdentifierSuppurative skin infections **Pseudomonas*, *Staphylococcus*, *Klebsiella*, *Proteus*, and *E*. *coli*Thirty-one patients were treated orally and locally for chronically infected skin ulcers with a 74% success rate[107,108,109]Acute postoperative empyema in chronic suppurative lung diseases **Staphylococcus*, *Streptococcus, E. coli*, *Proteus*, *Pseudomonas aeruginosa*, *Burkholderia dolosa*Phage–antibiotic combinations were used in the successful treatment of 45 patients[110,111,112]Complications due to bacterial infections in cancer patients **Staphylococcus* and *Pseudomonas*82% (65) successful treatment with phages compared to 61% (66) of patients treated with antibiotics [113]Recurrent subphrenic abscess *Antibiotic-resistant *E. coli*A single patient was successfully treated with phages after 33 days[114]Urinary tract infections (UTIs) **Staphylococcus*, *E. coli*, and *Proteus*Forty-six UTI patients were treated with phages with 92% making clinical improvements and 84% achieving bacterial clearance[115]Rhinitis, pharyngitis, dermatitis, and conjunctivitis **Staphylococcus*, *Streptococcus*, *E. coli*, *Proteus*, enterococci, and *P. aeruginosa*Patients were treated with phages (360), antibiotics (404), and phage–antibiotic combinations (576). Clinical improvements of 86%, 48%, and 83% across the treatment regimes, respectively[116]Cerebrospinal meningitis **K. pneumoniae*Successful treatment with orally administered phages in a newborn.[117]Bacterial diarrhea*E. coli*Orally administered coliphages showed no improvement in clinical outcome; some dysbiosis with streptococci was observed [118]Complicated or recurrent UTI patients with transurethral resection of the prostate *Enterococcus*, *E. coli*, streptococci, *P. mirabilis*, *P. aeruginosa*, staphylococciPatients with intravesical-administered pyophage cocktail, orally administered antibiotics, and a placebo bladder irrigation. Success rates of 18%, 28% and 37% were observed, respectively[85]Burn wounds*P. aeruginosa*Phages PP1131 showed no significant difference to standard of care antibiotics—patients treated with PP1131 were found to have phage-resistant *P. aeruginosa*[119]Prosthetic joint infections*S. aureus*, *S. epidermidis*, *S. lugdunensis*, *Streptococcus* sp., *E. faecium*, *E. faecalis*, *E. coli*, *P. aeruginosa*, and/or *K. pneumoniae*Phage treatment, with intraoperative and intravenous PhageBank™ bacteriophages, in conjunction with standard-of-care antibiotics/Debridement, Antibiotics, and Implant Retention (DAIR) procedures. Completion is predicted in 2024[87,91,120]Diabetic foot ulcers (DFUs)*Staphylococcus* spp., wound microbiomeUse of anti-staphylococcal phage gel (Intralytix Inc., Baltimore, MA, USA). Effect on bacterial microbiome of DFU wounds and patient outcomes. Trial was abandoned for funding reasons[121]Probiotic application for overall gut health*Bifidobacterium animalis* subsp. *lactis* BL04The use of bacteriophages (PreforPro) increased the survival and efficacy of probiotic bacteria administered vs. probiotics only vs. placebo[122]Phages preventing the acquisition of multidrug-resistant enterobacteria (PHAGE-BMR)*E. coli* or *K. pneumoniae* containing ESBL or carbapenemasesCollection of multidrug-resistant bacteria from patients in intensive care, subsequent search for presence and absence of phages in carriers/non-carriers. Currently active but of unknown status[123]Phage dynamics and influences during human gut microbiome establishment (METAKIDS)A broad range of bacteriophage and bacterial hosts Characterize phage and bacterial genomes, abundance, and variations during infant gut development. Terminated[124]Bacterial infection in cystic fibrosis patients*P. aeruginosa*A cocktail of 10 bacteriophages was used to reduce *Pseudomonas* presence after 6 and 24 h including sensitivity of isolates. Completed with no recorded outcomes[125]Prebiotic *Escherichia coli* and microbiotaCommercial coliphage cocktail effects on the microbiota and systemic inflammation. No disruption to microbiota and no effect on inflammatory markers [126]Venous leg ulcers*P*. *aeruginosa*, *S*. *aureus*, and *E. coli*Polyvalent phage preparation of 8 bacteriophages was assessed for their safety and efficacy. No available outcomes but the trial was completed[127]Lower urinary tract colonization*E. coli*Assess the safety, tolerability, pharmacokinetics, and pharmacodynamics of phage cocktail LBP-EC01 [128,129]Safety of topical phage solution intended for wound infections*S. aureus*Evaluating the safety and skin reactions to ascending doses of phages compared to the placebo [130,131]Abbreviation: ESBL—extended-spectrum β-lactamase. Those marked with a (*) originated in Poland and the Soviet Union.
viruses-16-00478-t002_Table 2Table 2Recent individual case studies of personalized phage therapy that impacted patients with multidrug-resistant infections.Infection(s)Bacterial SpeciesOutcome/CommentReferenceComplicated necrotizing pancreatitis*Acinetobacter baumannii*Clearance of *A. baumannii* and return to health using intravenously (IV) and percutaneously administered (9) phages screened from a phage bank [86]Bacteremia*P*. *aeruginosa*An IV-administered bacteriophage cocktail comprising two phages cleared the bacteremia, but the patient succumbed to other complications[132]Lung infection and transplant recipient *P*. *aeruginosa*An IV- and nebulizer-administered bacteriophage cocktail, AB-PA01 and Navy, with the patient recovering from pneumonia [111]Infection of left ventricular assist device*P*. *aeruginosa*Six-week IV-administered (3) phage cocktail; the patient was clear and then relapsed but a change in antibiotics led to recovery [133]Osteomyelitis*A*. *baumannii* and *K*. *pneumoniae*The patient developed postoperative infection with multidrug-resistant isolates. IV bacteriophage–antibiotic combination led to the patient’s full recovery without the need for amputation [133]UTIESBL *E. coli*Phage treatment with two phages over 23 days in conjunction with antibiotic treatment led to negative urine cultures and full recovery of the patient[132]CNS infection of a recovering trauma patient*A. baumannii*IV treatment with an *A. baumannii* phage for 8 days led to CSF cultures coming back negative for *A*. *baumannii* but positive for *K. pneumoniae* and *S*. *aureus*. The patient was declared brain dead and later announced deceased[132]Lung infection of cystic fibrosis patient*Achromobacter xylosoxidans*Cefiderocol and phage treatments were performed for 5 days followed by continuous phage therapy. The patient recovered and was discharged[132,134]Abbreviations: IV—intravenous; CNS—central nervous system; UTI—urinary tract infection; CSF—cerebrospinal fluid.

Trials conducted with phages selected for the treatment of bacterial infections have not all been successful. Subpopulations of bacteria develop resistance to phage attacks, which often occur at a rapid rate, rendering these strains immune to further phage infections. Results such as these raised serious doubts regarding the use of phages as therapeutic agents, as the rapid emergence of phage resistance nullifies this approach to the treatment of infections. The resistance mechanisms deployed by bacteria do alter virulence factors. This is especially true in a complex environment such as the gastrointestinal tract (GIT). This poses a major problem to the implementation of phage therapy and raises the question of whether beneficial gut microbiota may, over time, convert to disease-causing strains. Whatever the case may be, the emergence of resistance to phage treatment must be carefully monitored, as it may lead to spontaneous mutations and changes in the composition of phages. This, in turn, may alter the immune status of the patient, increase bacterial biofilm formation, and increase the possibility of pathogens developing acquired types of resistance, such as CRISPR (reviewed by Oechslin [135]). The widespread use of phages to treat infections may lead to an increase in phage-resistant bacterial pathogens, but this must be confirmed with further studies. The exchange of plasmids harboring genes encoding phage resistance is another concern that requires further research.

Phage therapy in its current form is used to treat persistent antibiotic-resistant bacterial infections and is more used as personalized medicine. A strategy that needs much more attention is the use of phage particles, instead of intact phages, combined with antibiotics. The possibility of using phage lysins and other phage-derived proteins is discussed in the next section.

### 6.2. Therapeutic Potential of Phage-Derived Proteins

Bacteriophages produce a range of enzymatically active proteins required for their adsorption, entry, and exit from their susceptible hosts. During the late phase of infection, bacteriophages produce endogenous lysins, allowing host lysis and the subsequent release of viral progeny (Figure 3). Lysins are part of a lysis cassette and rely on two other genes, namely holin and spanin, to help with the translocation of lysin across the cell membrane to peptidoglycan [7,8,9,10]. Gram-positive phage endolysins are usually composed of a two-domain structure, with an enzymatic catalytic domain at the N-terminal and a cell wall-binding domain at the C-terminal, separated by a short linker (Figure 4). Gram-negative phage endolysins are usually globular and displayed as a single enzymatic catalytic domain [11]. PlyPalA is an important lysin against *Paenibacillus larvae,* the causative agent of American foulbrood, which is detrimental to honeybees [136]. The activity of endolysins can vary and activity has been observed against sugars constituting the bacterial cell wall, i.e., they may be endo-β-N-acetylglucosaminidases or N-acetylmuramidases (lysozymes). Endopeptidases, which degrade protein moieties, and amidases such as N-acetylmuramoyl-L-alanine amidase, which degrade amide bonds between glycans and peptides, have also been reported. Lysins can also fall under a broader class of Cysteine Histidine-dependent Amidohydrolase/Peptidases (CHAPs) with one example of such observed in *Streptococcus pyogenes* producing a CHAP-like lysin that hydrolyzes the 1,4-β-glycosidic bonds between N-acetyl-*d*-glucosamine and N-acetylmuramic acid together in the peptidoglycan chain [137]. Contrary to the activity of the N-terminal, the C-terminal domain of lysins is usually involved in substrate binding and host specificity. Substrates include carbohydrates found in the cell wall of bacteria and the C-terminal is paramount for efficient cleavage of cell wall substrates [7]. These lysins form holes in the cell wall by hydrolyzing peptidoglycan, disrupting cell wall integrity and in turn hypertonic lysis. Although the impact of lysins on Gram-positive bacteria is promising, little activity is observed against Gram-negative bacteria, which could likely be due to the bioavailability of peptidoglycan being blocked by the Gram-negative cell envelope. Few endolysins are endogenously active in vivo, such as SPN9CC, PlyF307, and CfP1gp153. Lysins traverse the outer membrane with the help of external agents [138,139,140]. The mode of action of lysins has led to lysin-based medicinal applications, such as lysin–antibiotic combinations that can combat antibiotic-resistant bacteria. Djurkovic et al. [141] found various antibiotic combinations efficacious, such as gentamicin and penicillin, with a streptococcal phage lysin, CpI-1. The authors also found that a combination of penicillin and CpI-1 was highly active against previously penicillin-resistant strains. There have been recent successful results in a randomized controlled trial using an anti-staphylococcal lysin (exebacase) to treat bloodstream infections involving methicillin-resistant *S*. *aureus* [142]. They found that exebacase in conjunction with antibiotics proved more efficacious than antibiotics alone, and that treatment reduced hospitalization time by 4 days and readmission of patients by 48%.

Membrane-active agents such as Telavancin (a lipoglycopeptide antibiotic used to treat skin infections), Daptomycin (a cyclic lipopeptide antibiotic against Gram-positive bacteria, including methicillin-resistant *Staphylococcus aureus* and vancomycin-resistant enterococci), and Oritavancin (a glycopeptide antibiotic used to treat skin infections) have been used in medical applications (reviewed by Hurdle et al. [143]), and it would be interesting to see further development in their combined use with phage-produced lysins.

Another interesting approach is the artificial modification of endolysins by linking positively charged amino acids at either N- or C-termini. These modified endolysins, called artilysins, are attracted to the anionic outer membrane of bacteria and destabilize the integrity of the membrane to gain access to the peptidoglycan layer and cause cell lysis (reviewed by Carratalá et al. [144]). Briers et al. [145] fused two different endolysins (PVP-SE1gp146 and OBPgp279) to the nine-amino acid polycationic peptide PCNP (amino acid sequence KRKKRKKRK). Endolysins with a peptide fused to their N-terminus yielded slightly better antimicrobial activity compared to those with a fusion to their C-terminus [145]. Similar findings were reported when Wang et al. [146] fused endolysin JDlys to the cell-penetrating peptide CPP_Tat_ (amino acid sequence YGRKKRRQRRR). However, endolysins fused with their C-terminal to the peptide had no bactericidal activity [146]. In another study, Chen et al. [147] also generated two modified constructs by fusing the N- and C terminals of endolysin LysAB2 to the peptide cecropin (CecA) with amino acid sequence KWKLFKKI. In this case, the C-terminal modification was superior to the N-terminal modification, although the N-terminal fusion was slightly better compared to the native LysAB2 [147]. Concluded from these studies, the most optimal location for the fusion of lysin to a peptide may be influenced by other properties. It appears that hydrophobicity and amphiphilicity of the fused constructs also play a role in antibacterial activity [144]. Further research needs to be conducted on the fusion of lysins to antimicrobial peptides such as bacteriocins.

Contrary to endolysins, exolysins or phage-encoded depolymerases (Table 3) are usually found on phage tail fibers, tail spike proteins (TSPs), or the phage baseplate. Importantly, they cleave polysaccharides located on the bacterial cell envelope and are involved in host adsorption. Exolysins can be classified into two main classes, i.e., hydrolases and lyases, which act on a carbohydrate substrate such as capsule polysaccharides (CPSs), extracellular polysaccharide (EPS) matrices, and O-polysaccharides. Based on the substrate hydrolases act upon, they can be further subclassed into groups such as sialidases, rhamnosidases, levanases, xylanases, and dextranases. Many hydrolases rely on a water molecule to specifically cleave the O-glycosidic bonds between polysaccharide monomers [148]. Sialic acid capsules are used by several bacterial species including *E*. *coli* K1, *Haemophilus influenza*, *Streptococcus* spp., and *Campylobacter jejuni*. Capsules promote pathogenesis by improving adherence to surfaces, evasion of host immune responses, and biofilm formation, and acting as a nutrient source [149]. Phages encode endosialidases within their tail structures to overcome this carbohydrate barrier. Activity has been seen against a neuropathogenic *E*. *coli* K1 strain by the podovirus K1E, which encodes a hydrolytic tail spike protein that specifically binds and cleaves the K1 capsule [150]. An endosialidase, Endo92, from phage phi92 was capable of digesting K1 and K92 capsules of *E*. *coli* and is uniquely able to cleave both the α-2,8- and α-2,9-linkages of sialic acid [151]. Levanases are predominately found in bacterial species such as *Bacillus* and *Pseudomonas* and can hydrolyze the β-2,6-linked D-fructofuranosyl residues of levan [152,153]. Levan is an important structure in the development of a robust biofilm for *Bacillus* spp.; however, it is not a necessity. It plays a role in the stability of floating biofilms, can provide a nutritional reserve, and was found to be the major polysaccharide present in the EPS matrix [154]. Levanases have been found in several *B*. *subtilis* phages (SP10, ϕNIT1, and SPG24) and assist phages by exposing receptors [155]. Endorhamnosidase activity was first observed in *Salmonella* (ser.) Typhimurium phage P22, which degrades the O-antigen present on the LPS of S strains [156]. Specific cleavage by the P22 tail spike protein targets α-rhamnosyl 1-3 galactose linkages of the O-antigen, which is also seen in several other *Salmonella* phages [35]. Often, mutations in the LPS lead to insensitivity to certain bacteriophages but also contribute to less-virulent strains of bacteria [157]. An earlier study found that *Klebsiella* phages exhibit galactosidase or glucosidase activities, which cause the degradation of side chains present in CPSs [158].

Phage polysaccharide lyases cleave the 1,4 glycosidic bonds using a β-elimination mechanism. These enzymes appear to act on three types of polysaccharides including hyaluronate, alginate, and pectin, although not exclusively. Hyaluronidases first drew attention with several bacterial species producing them, and they were attributed to be a virulence factor for tissue permeability and pathogen invasion. It is thought that this same enzyme is used in streptococcal prophages to penetrate hyaluronic acid capsules, likely facilitating host entry. Alginate lyases can be mannuronate or guluronate lyases that degrade the two 1,4 glycosidic-linked monomers, α-L-guluronic acid and β-D-mannuronic acid, within alginate. Alginate provides structural integrity in brown algae but is also synthesized in *Pseudomonas* and *Azotobacter* species shown to contribute to biofilm formation. Alginate lyases are encoded in tail components of *Pseudomonas* and *Azotobacter* phages assisting penetration of phages across the acetylated poly(M)-rich EPS matrix allowing phages to bind to the cell envelope. Uropathogenic *E. coli* (UPEC) produce a capsular polysaccharide rich in colanic acid, which allows protection against hostile environments and promotes pathogenicity [159]. This negatively charged polymer contains glucose, galactose, fucose, and glucuronic acid and is upregulated in established biofilms [160]. There is evidence of phages overcoming this carbohydrate barrier; for example, Phi92 contains a colanidase tail spike protein which degrades colanic acid, allowing secondary tail spikes to degrade and/or bind to the cell envelope [151]. Lipases are rarely seen in phage genomes but are ubiquitous in nature. They have a broad specificity and often multifunctional properties. Phage lipases hydrolyze the carboxyl ester bonds of triacylglycerols, releasing organic acids and glycerol. The role of lipases in phages has yet to be elucidated [12,13]. There is some evidence that a lipase or esterase could be used to modify the O-antigen present on the LPS, preventing further phage infections [161].viruses-16-00478-t003_Table 3Table 3Bacteriophage-encoded depolymerases that contribute to host adsorption.Enzyme ClassPhage/EnzymePolymer SubstratesTargeted GeneraReferencesHydrolasesSialidasesPhi92Polysialic acid*E. coli* K1 and K92[151]K1E*E. coli* K5[162]K1F*E. coli* K1[163]LevanaseSP10Levan*Bacillus* spp.[155]SPG24RhamnosidaseSf6O-antigen LPS*Shigella flexneri*[156,164]P22Rhamnogalacturonan*Salmonella* (ser.) Typhimurium
CellulasesS6Cellulose*Erwinia amylovora*[165]PeptidasesCHAP_K_Pentaglycine cross-bridge peptidoglycan*Staphylococcus aureus*[166,167]phiNIT1Poly-γ-glutamate*Bacillus* spp.
LyasesHyaluronidasesProphagesHyaluronan*Streptococcus equi*[168,169]H4489A*Streptococcus pyogenes*
Alginate lyasesPT 6Alginic acid*P. aeruginosa*[14,65]AF*P. putida*
Pectin/pectate lyasesΦIPLA7Pectin **Staphylococcal* spp.[170]OthersColanidasePhi92Colanic acid*E. coli*[63]Lipases/triacylglycerol hydrolasesPhi3ST:2Carboxyl ester bonds **Cellulophaga* spp.[171]Tf*Pseudomonas* spp.* The exact role or substrate degraded by the phage-derived depolymerase is yet to be defined.

Researchers are looking into exploiting phage-derived depolymerases to make bacteria less virulent, assist in antibiotic treatment, act as prophylactics on medical devices, and improve immune responses to bacterial infections. There is strong evidence that phage depolymerases have potential as anti-biofilm agents; for example, phage alginate lyases can reduce biofilm formation of *P. aeruginosa* [14]. Alginate lyases can also improve antibiotic killing of mucoid *P. aeruginosa* [172]. Removal of the alginic acid EPS matrix is important for antibiotic efficacy as the EPS can block the bioavailability of gentamicin or tobramycin. Furthermore, the biofilm can directly bind aminoglycosides and cationic antibiotics [173,174]. Importantly, the removal of EPS-related virulence factors increases macrophage uptake of bacteria and exposure to immune complement, both contributing to the elimination of bacterial burden during infection [175,176]. Similar anti-virulent agents have been observed in *Klebsiella* phages producing capsular depolymerases that degrade CPS, reducing the virulence of carbapenem-resistant *K. pneumoniae* and exposing it to serum complement for effective killing [177]. Phage depolymerase–antibiotic combinations have been investigated, where phage depolymerase Dpo71 degraded *A*. *baumannii* CPS and reduced biofilm formation. Furthermore, the removal of CPS improved the antibacterial activity of colistin in a *Galleria mellonella* infection model [178]. Contrary to the success of Chen et al., a similar study performed using a CPS-degrading depolymerase, depoKP36, for *K. pneumoniae* noted that combination therapy did not improve antibiotic efficiency. Interestingly, no drug interference was observed with antibiotic–depoKP36 combinations. Removing CPS can improve phagocytosis and complement-mediated opsonization; therefore, further study should account for these immune responses when evaluating the use of phage depolymerases.

Bacteriophages have been investigated for their potential prophylactic use in lining medical equipment, especially catheters. Rice et al. [179] reported a pectate lyase domain in the tail of a *Proteus* phage that reduced the biofilm formation of *P*. *mirabilis.* The authors concluded that such a tail spike protein could be used for the treatment of catheter-associated UTIs (CAUTIs). Other studies have shown that catheters coated with bacteriophages can prevent biofilm formation [180]. Yet there are very few studies looking into depolymerases derived from phages on their own; most investigated the use of whole phage cocktails. Shahed-Al-Mahmud et al. [181] evaluated the anti-fouling capabilities of a phage tail spike protein against *A*. *baumannii* biofilms on catheter sections and found no inhibition of cell colonization. The therapeutic effect was further evaluated in a zebrafish model, which showed the tail spike protein increased the survivability of zebrafish by 80% when challenged with *A. baumannii*. This warrants further investigation into the use of phage-derived depolymerases as prophylactic coatings on medical devices. Clinical trials into the use of phage depolymerase cocktails in combination with antibiotics are important.

## 7. Limitations of Phage Therapy

A major limitation of the use of phages in the treatment of infections is the possibility of overstimulation of the immune system. Phage capsids, tails, and tail fibers are proteins and are thus recognized by the immune system. Some studies reported no major changes in the immune response of patients treated with phages [2]. These findings are supported by the observation that phages are phagocytosed within a few minutes after administration, at least according to tests conducted on animals and mammalian tissue cells [182]. In immunocompromised mice, phage T7 was eliminated within 60 min after injection [183]. In most cases, treatment with phage T7 did not increase pro-inflammatory cytokines and reactive oxygen species (ROS) and did not damage tissue [184]. Similar studies were reported with phage T4. The intraperitoneal injection of phage T4 head proteins in mice did not stimulate the production of interleukin (IL)-1a, IL-1b, IL-2, IL-6, IL-10, IL-12, p40/p70, interferon (IFN)-γ, tumor necrosis factor (TNF)-α, monocyte chemoattractant protein (MCP-1), monokine induced by gamma (MIG), RANTES, granulocyte colony-stimulating factor (GCSF), granulocyte–macrophage colony-stimulating factor (GM-CSF), and reactive oxygen species (ROS) [185]. These findings were supported by the findings of Hwang et al. [186]. Phages used to treat *Burkholderia cenocepacia* pulmonary infections in mice controlled the increase in bacterial cell numbers and did not stimulate the production of macrophage inflammatory protein 2 (MIP-2) and TNF-α [187]. Despite findings that the human immune system is not altered by phage therapy, antibodies against *E. coli* T4 phages were detected in more than 80% of patients who were not treated with phage T4 [188]. Findings such as these expose our limited knowledge of the interactions between phages and hosts.

## 8. Genomic Engineering of Phages

A research field that has been neglected in the past is the genetic engineering of phages. This, however, has changed over the last decade, and several strategies have been employed in using genetically engineered phages in antibacterial applications (phage therapy), disruption of biofilms, and delivery of antimicrobials. Genetic engineering has also been explored in the use of https://www.sciencedirect.com/topics/biochemistry-genetics-and-molecular-biology/endolysin (accessed on 5 February 2024) endolysins as antibacterial agents and altering the host range of phages. In eukaryotes, genetically engineered phages may be used to deliver genes and drugs to targeted cells. Genetically modified phages are also used in the development of vaccines and in tissue engineering. For further information, the reader is referred to the review by Hussain et al. [189]. The introduction of *rpsL* and *gyrA* in lysogenic phages increased the sensitivity of pathogens to streptomycin and nalidixic acid, respectively [190]. This approach may be used in the treatment of methicillin-resistant *S. aureus*, as shown in the treatment of skin infections [191] and bacterial infections associated with wounds [189,192], burns [119,190,191,193], and diabetic leg and foot ulcers [192,194,195,196]. Phages have also been used in the treatment of wound sepsis caused by multidrug-resistant *P. aeruginosa* [197]. Positive results obtained with phages in the treatment of bacterial skin infections were questioned by data generated from a clinical trial in which phage treatment was compared with sulfadiazine cream [119]. Concluded from this study, the phages used in the treatment were unstable, which resulted in patients receiving 1000-fold to 10,000-fold fewer phages than initially prescribed.

Genome engineering of phages may be performed in several ways and includes recombination between phage DNA and plasmids [198], the use of shuttle plasmids [199], cloning of specific genes [200], recombineering [201,202], CRISPR-Cas selection [202], and a combination of recombineering and CRISPR-mediated counter-selection [203]. For a review of the advances of genetically engineered phages, the reader is referred to Pires et al. [204].

Although the oral administration of phages prevented cholera [205], randomized controls were not included. In other studies, inconclusive results were obtained when patients with *E. coli* diarrhea were treated with phage cocktails [118,206,207,208]. An increase in intestinal levels of *Streptococcus gallolyticus* and *Streptococcus salivarius* was recorded [118], which may indicate an imbalance in gut microbiota. The authors ascribed these changes to the fact that only 60% of the 120 patients who were enrolled in the trial showed *E. coli* in their stool and that the population of phages used in the treatment did not increase in the GIT.

Phages may be used to control biofilm formation, as shown by the ability of some phages to produce depolymerases [209]. Recently, three depolymerases were identified in the genome of *Klebsiella pneumoniae* phages [210]. The depolymerases destroyed *K. pneumoniae* capsule serotypes K7, K20, and K27, and revealed promising results in vivo in a *Mus musculus* survival study. Majkowska-Skrobek et al. [211] showed that a phage depolymerase sensitized *K. pneumoniae* against serum-mediated killing and phagocytosis. Similar findings were reported for phages used in the treatment of multidrug-resistant *Acinetobacter baumannii* [175,178], *E. coli* [212,213], *P. aeruginosa* [214], and *Proteus mirabilis* [172]. Lu and Collins [200,206] engineered a phage to express biofilm-degrading enzymes and destroy an *E. coli* biofilm. The engineered phage reduced bacterial biofilm cell counts by approximately 99.9%. In-depth studies on depolymerases produced by phages may lead to the development of vaccines against capsules of antibiotic-resistant bacterial strains.

## 9. Diagnostic Potential of Phages and Phage-Derived Proteins

The diagnostic potential of phage-derived proteins has been investigated. A bioinformatic tool, Kaptive, was developed to rapidly identify capsule and lipopolysaccharide (K and O) types of *K. pneumoniae* and *A. baumannii* [215]. Recently, *Vibrio parahaemolyticus* was added to this database [216]. Rapid capsule-typing tools have several advantages over traditional capsule typing which includes serological reactivity assays or polymerase chain reaction-based sequencing [217]. Serotyping was first introduced in 1926 and has been used ever since to type capsules. For example, capsule serotyping has identified 77 capsule types (K-types) of *K. pneumoniae*, while genotyping has found 134 KL (K-locus) types [64,217]. Although important, several limitations of serotyping and genotyping have been reported. Limitations include the inability to distinguish between genotypically similar KL types. It is also an expensive and tedious process to produce antisera. Furthermore, modifications in capsular polysaccharides cause capsular variation, to the extent that singular capsule types were described for *Streptococcus pneumoniae* and *E. coli* [218].

Alternative methods could include using whole phages or phage-derived proteins for capsule typing. Success has been seen in this regard, where whole phages and phage-derived proteins have been used not only for capsule and LPS typing but also for other diagnostic purposes. Li et al. [219] discovered a novel phage, *Klebsiella* phage SH-KP152410, that specifically recognized the KL64 capsule type of clinical *K. pneumoniae* strains. The authors also showed that the depolymerase (K64-ORF41) could reliably type K64 (serotyping) and KL64 *K. pneumoniae* strains in agreement with genotyping and serotyping. They also demonstrated one of the limitations of genotyping where clinical strains were erroneously typed because an insertion occurred in the *wcaJ* gene sequence, which encodes WcaJ, responsible for the initiation of capsule biosynthesis. Park and Park [220] identified an O-antigen (located in the LPS)-active depolymerase, Dpo10, in the genome of an *Escherichia* siphophage. The authors tested the specific typing capabilities of Dpo10 and showed that the depolymerase only acted upon *E. coli* O157:H7 strains. This study further substantiates the viability of applying phage depolymerases to specifically type bacterial CPSs and LPSs. Moreover, several other studies have proven the feasibility of phages and phage-derived proteins in fundamental studies regarding bacterial defense and virulence mechanisms. Dunstan et al. [221] proved that depolymerase can be used to characterize the monomers in the capsule of *K. pneumoniae*. This fundamental information can be used to generate capsule-specific vaccines. Other articles have used whole phages in combination with nuclear magnetic resonance and mass spectrometry to identify the constituents and structures of capsules.

Reliable, sensitive, and fast detection of nosocomial pathogens such as *E. coli*, *K. pneumoniae*, *P. mirabilis*, and *A. baumannii* is urgently required [222]. Therefore, multiple efforts have focused on creating point-of-care biosensor devices that specifically recognize pathogens. Chen et al. [222] created a biosensor that could specifically and reliably (100% recognition) detect *A. baumannii* strains (n = 77). The authors used a heterologously expressed phage-derived receptor-binding protein, called Gp50, which showed higher specificity than the whole phage (100% vs. 27.3%). They were also able to quantify *A. baumannii* in colony-forming units per mL and observed 100% recognition even in complex sample matrices.

## 10. Conclusions

Intact bacteriophages and phage-derived proteins have various advantageous properties when used as therapeutic agents and diagnostic probes. Therapeutically, whole-phage preparations have shown promise but with a more personalized medicine approach. This approach requires a rapid pipeline for the isolation, purification, and characterizing of therapeutically suitable phages. Screening for phage resistance must include studies on phage-resistant mutant populations, the antibiotic susceptibility of bacteria, hypovirulence factors, and susceptibility to immune responses. The concept of using phage-derived proteins such as endolysins and depolymerases, alone and in combination with conventional antibiotics, to treat previously antibiotic-resistant bacteria requires further research. Antibiotic treatment could be improved by the simultaneous use of depolymerases that could decapsulate or degrade biofilms, thereby increasing antibiotic bioavailability. This approach could be the answer to the treatment of chronic infections, especially since the development of new antibiotics is an expensive and time-consuming process. Although conceptually promising, the widespread implementation of phage therapy in routine clinical practices is restricted by the lack of safety and efficacy data collected according to clinical trial standards and regulations. Care should be taken when results are interpreted, as clinical studies differ in design and aim, e.g., treatment with single phages vs. phage cocktails, routes of administration (intravenous, oral, local, or combined), and treatment in combination with antibiotics.

Phage therapy might be highly efficacious in eradicating pathogens in well-defined and circumscribed infected niches, particularly if used in combination with antibiotics. The biggest advantage of phage therapy over the use of antibiotics lies in their rapid killing abilities and that they can self-replicate at the site of infection. Phages may increase antibiotic susceptibility in specific cases. If used with antibiotics, the emergence of phage-resistant pathogens may be kept under control. Although these are promising ways of treating bacterial infections, thorough and systematic evaluation processes must be in place to avoid the development of uncontrollable spontaneous bacterial resistance.

Sequencing of bacterial and phage genomes is important to identify the defense and anti-defense systems of pathogens. This will also help us select phages, have a better understanding of how bacterial defense systems affect phage therapy, and learn more about the anti-defense strategies employed by phages to counteract bacterial defenses. Clinical trials need to be carefully designed to evaluate the efficacy of phage therapy. More research needs to be conducted on the preparation of phage formulations, whether used alone or in combination with antibiotics, to prevent the development of bacterial-resistant strains as encountered with the use of antibiotics. More in-depth knowledge is required to understand the co-evolution between phages and bacteria before phage treatment can be declared an answer to the treatment of a broader spectrum of infections.

## Figures and Tables

**Figure 1 viruses-16-00478-f001:**
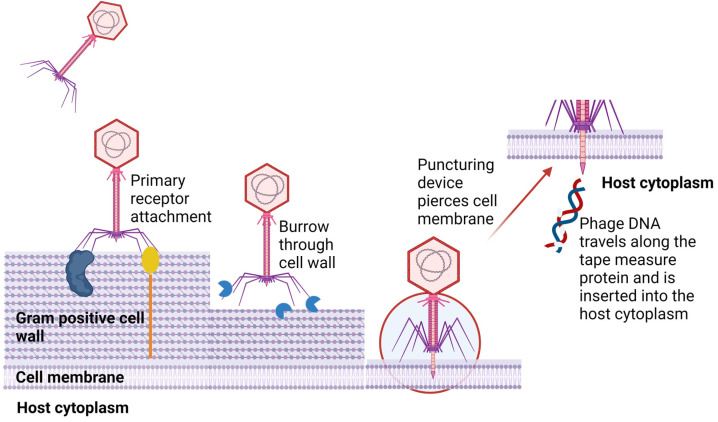
A generalized representation of a tailed phage with primary receptors on long tail fibers. Irreversible binding of the central distal tail components punctures the host cell membrane, and the viral genome is released [28] Created using Biorender.com (accessed on 5 February 2024).

**Figure 2 viruses-16-00478-f002:**
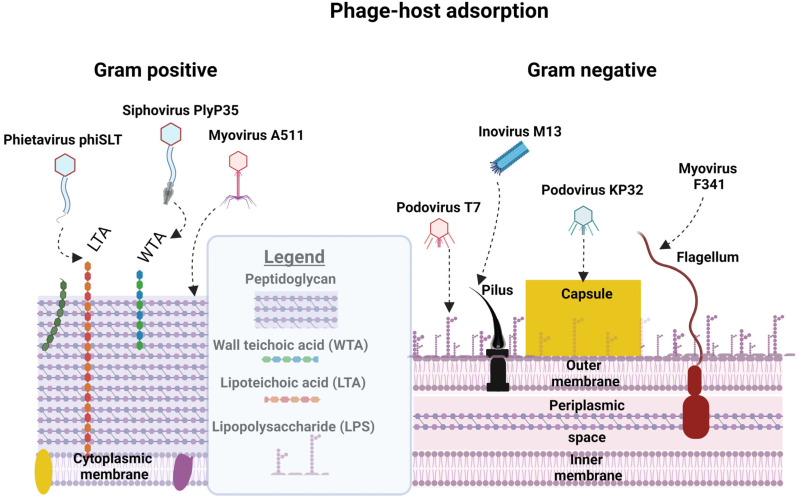
Phages use different cell structures as attachment sites and for entering the host’s cytoplasmic membrane. Identified phage receptors for Gram-positive bacterial cells include peptidoglycan (e.g., A511), WTA (*Listeria* phage Plyp35), lipoteichoic acids (LTAs, staphylococcal phage phi SLT), and flagella (*Campylobacter* phage F341). Gram-negative hosts can be targeted via their lipopolysaccharides (LPSs, *E. coli* Phage T7), pili (*E. coli* Phage M13), and capsules (*Klebsiella* phage KP32). Information was obtained from Dunne et al. [31]. Created using Biorender.com (accessed on 5 February 2024).

**Figure 3 viruses-16-00478-f003:**
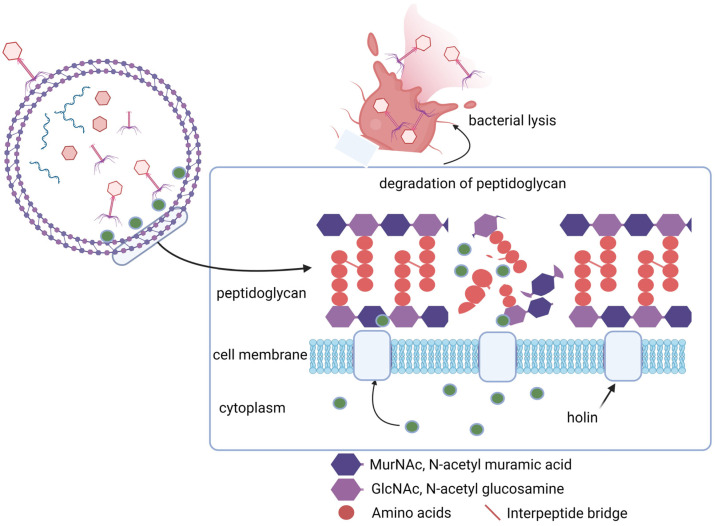
The role of endolysins (green circles) and holins in the process of phage infection (as an example, in a Gram-positive bacterium). After replication inside the bacterial cell, progeny phages use a lytic system, including endolysins and holins, to destroy the cell wall from the inside of the bacterium and release the assembled phage virions (created using Biorender.com, 5 February 2024).

**Figure 4 viruses-16-00478-f004:**
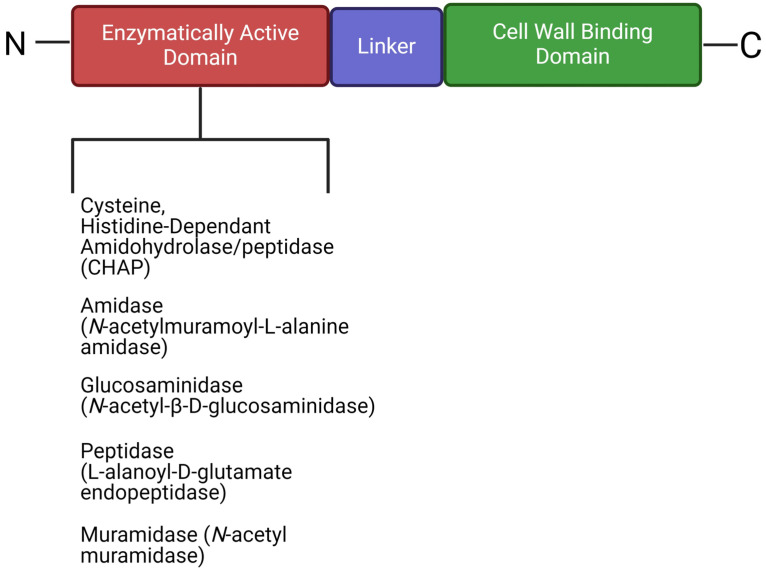
The generalized structure of Gram-positive phage endolysins containing an N-terminal enzymatic domain and a cell wall-binding domain on the C-terminal (created using Biorender.com, 5 February 2024).

## Data Availability

Not applicable.

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
