# Peer review of "Bacteriophage–Host Interactions and the Therapeutic Potential of Bacteriophages"

_viruses, 2024, doi:10.3390/v16030478_

Round 1

Reviewer 1 Report

Comments and Suggestions for Authors

Dicks and Vermeulen review a large body of literature dealing with the use of bacteriophages and/or phage-derived proteins to fight pathogenic bacteria and the infections caused by them. The entire manuscript reads rather uninspired. The authors do contribute few original thoughts that would distinguish their text from previous reviews in the field and this includes figures and tables. Also, a few mistakes in the text, as listed below, throw some shade on the expertise of the authors. On a positive note, the topic addressed by the authors is important and deserves to be kept in the spotlight of the scientific discussion.

Specific comments: 

Line 39: “…produced during the final phase of bacterial infection.” The term “bacterial infection” in this context is not correct. Do the authors mean in “the final phase of a bacteriophage infecting a bacterial cell”?

Line 40: Typo should read “spanin” not “spannin”

Line 86: “… the surface of the host cell membrane [23].” Better delete the term “membrane” from this sentence, otherwise this generalized statement would exclude all the Gram-positive bacteria.

Line 137: “polysaccharide layer” or “exopolysaccharide layer” not “depolysaccharide layer”.

Figure 2: Siphoviruses do have long tailfibers. It was a historical artifact that early pictures of phage lambda were taken of a mutant lacking tailfibers.

Figure 2: Campylobacter is a Gram-negative bacterium, accordingly F341 and the flagella complex should move to the righthand side of the figure.

Line 160: “… the phage proceeds to enter the host’s cytoplasm”. Unfortunate formulation, the phage particle stays outside, so it is not really the phage that proceeds to enter the host, but rather the genetic material of the phage.

Lines 171-172: “Siphoviridae phage”. Old nomenclature, the authors correctly claimed to be disused. Better “Siphovirus phage”.

Line 179: “Podoviridae burrows” as above, old nomenclature, better “Podoviruses burrow”.

Lines 185-187: “Tectivirus PRD1 is an unusual phage in that it directly fuses with the host membrane and, as it is non-enveloped, and has a capsid that surrounds an inner membrane.” In this form the entire sentence does not make any sense, even without the duplication of “and”.

Line 235: Typo should read “pyogenes” not “pyrogenes”

Lines 257-259: “Much more research is required to have a complete understanding of how phages develop resistance, especially in the context of the human body.” Really “phages develop resistance”, not rather bacteria develop resistance to phages!?

Line 361: “… nine to β-lactamase.” “β-lactamase is a protein” the authors probably mean “ … nine to β-lactam antibiotics”!?

Line 402: As in line 40, “spanin” not “spannin”

Line 407: “… main [11. PlyPalA …” Incomplete citation?

Line 419: “These lysins form pores in the cell wall …” Is “pore” the right term in this context? Would not “hole” be more appropriate?

Line 478 and line 482 and table 3: "colanic acid" not "colonic acid"

Line 607: "phage-derived" rather than "phamolge-derived"?

Author Response

Dicks and Vermeulen review a large body of literature dealing with the use of bacteriophages and/or phage-derived proteins to fight pathogenic bacteria and the infections caused by them. The entire manuscript reads rather uninspired. The authors do contribute few original thoughts that would distinguish their text from previous reviews in the field and this includes figures and tables. Also, a few mistakes in the text, as listed below, throw some shade on the expertise of the authors. On a positive note, the topic addressed by the authors is important and deserves to be kept in the spotlight of the scientific discussion.

Answer: Additional information has been added to the paper and mistakes have been corrected, as also explained below.

Specific comments:

Line 39: “…produced during the final phase of bacterial infection.” The term “bacterial infection” in this context is not correct. Do the authors mean in “the final phase of a bacteriophage infecting a bacterial cell”?

Answer: The sentence now reads “….the final phase of a bacteriophage infecting a bacterial cell” (line 39).

Line 40: Typo should read “spanin” not “spannin”

Answer: Corrected (line 40).

Line 86: “… the surface of the host cell membrane [23].” Better delete the term “membrane” from this sentence, otherwise this generalized statement would exclude all the Gram-positive bacteria.

Answer: The sentence now reads “…..the surface of the host cell [29]” (line 101).

Line 137: “polysaccharide layer” or “exopolysaccharide layer” not “depolysaccharide layer”.

Answer: Corrected to “polysaccharide layer” (line 152).”

Figure 2: Siphoviruses do have long tailfibers. It was a historical artifact that early pictures of phage lambda were taken of a mutant lacking tailfibers.

Answer: This is correct and the image for Siphoviruses in Fig. 2 has been corrected.

Figure 2: Campylobacter is a Gram-negative bacterium, accordingly F341 and the flagella complex should move to the righthand side of the figure.

Answer: Correct, thank you. Figure 2 has been corrected by placing Myovirus on the right of the schematic drawing.

Line 160: “… the phage proceeds to enter the host’s cytoplasm”. Unfortunate formulation, the phage particle stays outside, so it is not really the phage that proceeds to enter the host, but rather the genetic material of the phage.

Answer:  The sentence now reads “….the genetic material of the phage enters the host’s cytoplasm.” (line 163).

Lines 171-172: “Siphoviridae phage”. Old nomenclature, the authors correctly claimed to be disused. Better “Siphovirus phage”.

Answer: Corrected (line 172)

Line 179: “Podoviridae burrows” as above, old nomenclature, better “Podoviruses burrow”.

Answer: Corrected (line 179)

Lines 185-187: “Tectivirus PRD1 is an unusual phage in that it directly fuses with the host membrane and, as it is non-enveloped, and has a capsid that surrounds an inner membrane.” In this form the entire sentence does not make any sense, even without the duplication of “and”.

Answer: The sentence has been amended to read: “Tectivirus PRD1 is an unusual phage in that with the help of nonstructural virion-encoded assembly factors and coat-forming proteins, a virus-specific lipoprotein membrane obtains an outer protein shell leading to the formation of a procapsid, which is translocated to the interior of the cell.” (lines 195-198).

Line 235: Typo should read “pyogenes” not “pyrogenes”

Answer: Corrected (line 230).

Lines 257-259: “Much more research is required to have a complete understanding of how phages develop resistance, especially in the context of the human body.” Really “phages develop resistance”, not rather bacteria develop resistance to phages!?

Answer: The sentence now reads: “Much more research is required to have a complete understanding of how bacteria develop resistance to phages, especially in the context of the human body.” (lines 252-254).

Line 361: “… nine to β-lactamase.” “β-lactamase is a protein” the authors probably mean “ … nine to β-lactam antibiotics”!?

Answer: Corrected to read “and nine to β-lactam antibiotics” (line 357).

Line 402: As in line 40, “spanin” not “spannin”

Answer: Corrected (line 403).

Line 407: “… main [11. PlyPalA …” Incomplete citation?

Answer: Corrected (line 408).

Line 419: “These lysins form pores in the cell wall …” Is “pore” the right term in this context? Would not “hole” be more appropriate?

Answer: Now reads: “These lysins form holes in the cell wall…..” (line 420)

Line 478 and line 482 and table 3: "colanic acid" not "colonic acid"

Answer: Corrected (line 510 and in Table 3).

Line 607: "phage-derived" rather than "phamolge-derived"?

Answer: Corrected (line 639)

Reviewer 2 Report

Comments and Suggestions for Authors

This review article from Dicks and Vermeulen highlights research  on bacteriophage-host  interactions and therapeutic potential of bacteriophages. The review might be helpful for readers interested in these topics. However, I have the following comments to be addressed.

Major points:

-Lines 64-65. It would be helpful for the reader to see references to the previous classification.

-Lines 66-71. The classification of bacteriophages should be presented according to the latest Master Species List.

- The authors should check the correct use of the term “virome”  throughout the text. For example, lines 130-131. Using this term in a phrase “…tactics used to inject the virome” may confuse the reader.

-The usage of names of abolished groups may confuse the reader.

For example, line 163 “Myoviridae…”; lines 171-172  “Phage T5, a Siphoviridae phage…”; line 179 “…Podoviridae… “.

-Lines 261-264. The authors should clarify in the MS text which antiviral defense systems are innate and which are adaptive.

-Lines 388-389. “...(reviewed by Oechslin)”. References to relevant works are required.

-Lines 423 – 424. “Lysins traverse the outer membrane with the help of external agents [141-143]“. It may be helpful to the reader to see a brief listing of agents that disrupt the permeability of the bacterial cell membrane that are used in medical practice.

-The authors discuss the successful usage of endolysins in combination with antibiotics. It would also be useful to see a short discussion of such an interesting approach as the development of chimeric constructs – artilysins, which are endolysins fused with cationic peptides.

-A figure with the structure of peptidoglycan showing the position of hydrolysis by lytic enzymes would facilitate the reader's comprehension of the text. This figure would be appropriate in the section “Therapeutic Potential of Phage-derived Proteins”.

Minor points:

-Line 65. The word “Caudovirales” should be written in italics.

-Line 177. If you want to use the name “podoviruses”, it is better to write it with a small letter.

-Line 185. “...Tectivirus…” This is not the official name of this virus and if you use it, it should be written in small letters and a regular font.

-Table 1. Line, where reference [126] is mentioned. There may be an extra full stop in the phrase  “A broad range of bacteriophage and bacterial hosts”.

-Line 407. Reference [11] is missing a parenthesis.

-Line 432. There probably should be “hospitalization” instead of “hoptizilization”.

-Line 607. There probably should be “phage-derived proteins” instead of “phamolge-derived proteins”.

Author Response

This review article from Dicks and Vermeulen highlights research  on bacteriophage-host  interactions and therapeutic potential of bacteriophages. The review might be helpful for readers interested in these topics. However, I have the following comments to be addressed.

Major points:

-Lines 64-65. It would be helpful for the reader to see references to the previous classification.

Answer: References 16 and 17 have been added (line 66).

-Lines 66-71. The classification of bacteriophages should be presented according to the latest Master Species List.

Answer: The classification is presented according to Master List 37 (as mentioned in the text: lines 66-71) and names have been corrected throughout the paper

- The authors should check the correct use of the term “virome”  throughout the text. For example, lines 130-131. Using this term in a phrase “…tactics used to inject the virome” may confuse the reader.

Answer: This has been corrected to “virus genome” where applicable (lines 146, 353 and 361).

-The usage of names of abolished groups may confuse the reader.

For example, line 163 “Myoviridae…”; lines 171-172  “Phage T5, a Siphoviridae phage…”; line 179 “…Podoviridae… “.

Answer: These have been corrected to Myoviruses (line 165) and Siphovirus (line 172).

-Lines 261-264. The authors should clarify in the MS text which antiviral defense systems are innate and which are adaptive.

Answer: The following sentence was added: “Adaptive immune systems such as CRISPR-cas target and degrade nucleic acids derived from bacteriophages and other foreign genetic elements, whereas innate immune systems rely more on restriction modifications, DNA degradation systems, and abortive infection [70].” (lines 258-261).

-Lines 388-389. “...(reviewed by Oechslin)”. References to relevant works are required.

Answer: A reference was added: “…..(reviewed by Oechslin [140]).” (line 390).

-Lines 423 – 424. “Lysins traverse the outer membrane with the help of external agents [141-143]“. It may be helpful to the reader to see a brief listing of agents that disrupt the permeability of the bacterial cell membrane that are used in medical practice.

Answer: A section has been added to give examples of agents that disrupt the permeability of the bacterial cell membrane and are used in medical practice (lines 444-449).

-The authors discuss the successful usage of endolysins in combination with antibiotics. It would also be useful to see a short discussion of such an interesting approach as the development of chimeric constructs – artilysins, which are endolysins fused with cationic peptides.

Answer: A section on artilysins has been added, with reference to papers (lines 450-469).

-A figure with the structure of peptidoglycan showing the position of hydrolysis by lytic enzymes would facilitate the reader's comprehension of the text. This figure would be appropriate in the section “Therapeutic Potential of Phage-derived Proteins”.

Answer: Please see the newly added figure (Fig. 3).

Minor points:

-Line 65. The word “Caudovirales” should be written in italics.

Answer: Corrected (line 65).

-Line 177. If you want to use the name “podoviruses”, it is better to write it with a small letter.

Answer: Corrected (line 177)

-Line 185. “...Tectivirus…” This is not the official name of this virus and if you use it, it should be written in small letters and a regular font.

Answer: Corrected (line 195).

-Table 1. Line, where reference [126] is mentioned. There may be an extra full stop in the phrase  “A broad range of bacteriophage and bacterial hosts”.

Answer: Corrected (full-stop has been deleted). With the changes made to the paper, this is now opposite reference 127 in the table.

-Line 407. Reference [11] is missing a parenthesis.

Answer: Corrected (line 408).

-Line 432. There probably should be “hospitalization” instead of “hoptizilization”.

Answer: Corrected (line 433).

-Line 607. There probably should be “phage-derived proteins” instead of “phamolge-derived proteins”.

Answer: Corrected (line 639).

Reviewer 3 Report

Comments and Suggestions for Authors

It was interesting and informative to read the MS entitled “Bacteriophage-host interactions and the therapeutic potential of bacteriophages” by L.M.T. Dicks and W. Vermeulen, which was submitted to MDPI Journal “Viruses”.

The current broadly faced problem is related to over-usage of antibiotics that were the main medicine for quite long period of time for the patients suffering from bacterial infections. Bacteria that infect humans (specifically in hospitals) and animals are typically present in large quantities in these organisms, and mutations of bacterial DNA generate many changes in their genetic allowing to develop defence mechanisms against antibiotics. The bacteria, that survived such treatment by antibiotics, propagate intensively and are not affected anymore by classical medications. Nowadays majority of antibiotics are not effective against their targeted bacteria.

Authors of this review have outlined recent reports on usage bacteriophages for treatments of bacterial infections, and described some problems that were encountered during different approaches at treatments of deceases caused by bacteria.  At the present time there are newly developed approaches where the usage of phages provides more effective treatment of these diseases. Nonetheless, we do not have yet a well-established reliable recipes that would allow an effectual medical treatment with known (or just a few of them) phages, since the bacteria are still capable to develop resistance to phages.  The authors have described some approaches used for the treatments of bacterial infections, some combinations of the current methods, and provided some examples of how they have been used in particular cases. The authors have provided interesting and useful overview of latest achievements reported during the latest years.

Below are comments related to some corrections that would be good to make.  

Line 30. ”… they can also infect several strains within a (?)  species, multiple species and/or multiple genera”. The sentence is rather repetitive and has to be rewritten in a clearer stile.

Lines 52-52. “Some phages produce endosialidases at their tail structures to degrade polysaccharide barriers.” Please rephrase the sentence.  The tail structures do not produce any proteins, they comprise proteins the functions of which depend on the phase of the phage activity.

Lines 63-83, Section “Classification of Bacteriophages”. The paragraph is rather confusing, since while the authors have made very brief introduction to the new classification of the phages, the authors themselves have used the old classification based on the morphology throughout the entire review.  It was not clear what was the point to include this section into this review, when the links between the old and new system of the classification were not done, and it was not used in the review. It does not add anything to the main topic of the review.

Lines 203-218.  Section 4. “Phage-host interactions”. It is a general information related to the overall description of interactions of phages with the bacterial hosts. It should be moved before section 3, where the authors describe these interactions in more details.

Line 83. “Reversible binding allows desorption and the re-infection of another host.” The sentence has to be modified to indicate that a phage will try to be adsorbed to another bacterium cell of the same host culture (not another type of the host, as it sounds in this sentence). It will be not re-infection, but only infection of the other bacterial cell.

Line 94. “… transfection of the host” . Should it be “… transfection into the host.”?

Lines 160-161. “…through a variety of different measures, depending on the phage’s morphology”. Please explain what do you mean as “different measures”? More information is needed.

Lines 169-170. “… the precise role of the tape measure protein is unknown; it is believed to aid movement across the host’s thick cell envelope”. It is as well not clear sentence. What do the authors mean as “the precise role” of a protein? Functions of many phage proteins depend on the step of the phage activity. During the self-assembly process of a phage the tape measure proteins define the length of the phage tail, if the phage became adsorbed to the host bacterium it helps to dissolve cell envelope.

The authors should carefully read their paper again and the appropriate corrections, since possibly there are more other small inconsistences in sentences.

Possibly it would be good to have some sort of statistics that will show which type of bacteria were most successfully treated by phages and were thebphage cocktails or specifically used (designed) phages.  We still have to know more how the specific lytic phages should be selected, modified or even created.  Such statistic would help to analyse and find such phages and to determine their efficacy. However, possibly it should be another review.

Comments on the Quality of English Language

/

Author Response

It was interesting and informative to read the MS entitled “Bacteriophage-host interactions and the therapeutic potential of bacteriophages” by L.M.T. Dicks and W. Vermeulen, which was submitted to MDPI Journal “Viruses”.

The current broadly faced problem is related to over-usage of antibiotics that were the main medicine for quite long period of time for the patients suffering from bacterial infections. Bacteria that infect humans (specifically in hospitals) and animals are typically present in large quantities in these organisms, and mutations of bacterial DNA generate many changes in their genetic allowing to develop defence mechanisms against antibiotics. The bacteria, that survived such treatment by antibiotics, propagate intensively and are not affected anymore by classical medications. Nowadays majority of antibiotics are not effective against their targeted bacteria.

Authors of this review have outlined recent reports on usage bacteriophages for treatments of bacterial infections, and described some problems that were encountered during different approaches at treatments of deceases caused by bacteria.  At the present time there are newly developed approaches where the usage of phages provides more effective treatment of these diseases. Nonetheless, we do not have yet a well-established reliable recipes that would allow an effectual medical treatment with known (or just a few of them) phages, since the bacteria are still capable to develop resistance to phages.  The authors have described some approaches used for the treatments of bacterial infections, some combinations of the current methods, and provided some examples of how they have been used in particular cases. The authors have provided interesting and useful overview of latest achievements reported during the latest years.

Answer: Thank you.

Below are comments related to some corrections that would be good to make.  

Line 30. ”… they can also infect several strains within a (?)  species, multiple species and/or multiple genera”. The sentence is rather repetitive and has to be rewritten in a clearer stile.

Answer: Now corrected to read “….but they can also infect several strains within a species, and multiple genera” (line 30).

Lines 52-52. “Some phages produce endosialidases at their tail structures to degrade polysaccharide barriers.” Please rephrase the sentence.  The tail structures do not produce any proteins, they comprise proteins the functions of which depend on the phase of the phage activity.

Answer: Now reads: “Some phages have endosialidases at their tail structures to degrade polysaccharide barriers” (lines 52 and 53).

Lines 63-83, Section “Classification of Bacteriophages”. The paragraph is rather confusing, since while the authors have made very brief introduction to the new classification of the phages, the authors themselves have used the old classification based on the morphology throughout the entire review.  It was not clear what was the point to include this section into this review, when the links between the old and new system of the classification were not done, and it was not used in the review. It does not add anything to the main topic of the review.

Answer: We would like to keep the classification section (section 2) to emphasize the changes in nomenclature.  We appreciate the comment and acknowledge that the classification of bacteriophages is not the main focus of the paper.  The old names have been replaced with new names throughout the paper.

Lines 203-218.  Section 4. “Phage-host interactions”. It is a general information related to the overall description of interactions of phages with the bacterial hosts. It should be moved before section 3, where the authors describe these interactions in more details.

Answer: This section has now been moved to the first part of section 3, as suggested (inserted in lines 85 to 99).

Line 83. “Reversible binding allows desorption and the re-infection of another host.” The sentence has to be modified to indicate that a phage will try to be adsorbed to another bacterium cell of the same host culture (not another type of the host, as it sounds in this sentence). It will be not re-infection, but only infection of the other bacterial cell.

Answer: The sentence now reads “Reversible binding allows desorption and infection of cells from the same strain in the culture” (line 104).

Line 94. “… transfection of the host” . Should it be “… transfection into the host.”?

Answer: Changed to “…transfection into the host”.

Lines 160-161. “…through a variety of different measures, depending on the phage’s morphology”. Please explain what do you mean as “different measures”? More information is needed.

Answer: This sentence is not necessary and has been deleted (line 171). The explanations before and after the now deleted sentence cover the subject.

Lines 169-170. “… the precise role of the tape measure protein is unknown; it is believed to aid movement across the host’s thick cell envelope”. It is as well not clear sentence. What do the authors mean as “the precise role” of a protein? Functions of many phage proteins depend on the step of the phage activity. During the self-assembly process of a phage the tape measure proteins define the length of the phage tail, if the phage became adsorbed to the host bacterium it helps to dissolve cell envelope.

Answer: The sentence now reads: “This process is similar in Gram-positive bacteria, except that phages use specialized tape measure proteins and the opening of a proximal plug that joins the capsid with the tail and then releases the phage genome” (lines 169-171).

The authors should carefully read their paper again and the appropriate corrections, since possibly there are more other small inconsistences in sentences.

Answer: The paper has been screened for other mistakes.

Possibly it would be good to have some sort of statistics that will show which type of bacteria were most successfully treated by phages and were thebphage cocktails or specifically used (designed) phages.  We still have to know more how the specific lytic phages should be selected, modified or even created.  Such statistic would help to analyse and find such phages and to determine their efficacy. However, possibly it should be another review.

Answer: This is a good point but we believe an interesting subject for a separate review.

Round 2

Reviewer 2 Report

Comments and Suggestions for Authors

The authors took into account a number of my comments. In general, it could be a “Minor revision”. However, I consider the issue related to the classification of bacteriophages to be important.

-The authors provide a classification according to the Master species list 37. However, the Master species list 38 was released quite a long time ago. The classification of bacteriophages in MSL 37 and MSL 38 has considerable differences. It would be much more useful for the reader to have up-to-date information. In addition, the authors write in MS (lines 69-71) that in the classification of bacterial viruses according to the MSL 37 there are only four orders. This is not true since these four orders only relate to the class Caudoviricetes. Other taxonomic groups of bacterial viruses also have orders. The authors should very carefully revise the MS text regarding taxonomy and provide up-to-date data.

Minor  points:

-line 73. There is no need to underline the family name.

-line 172. the word “Siphovirus” should be written with a small letter in a normal font.

Author Response

Dear Reviewer

Thank you for pointing out the outdated information regarding the classification of bacteriophages. We have now updated the review with information from the Master Species List 38 and trust you will find this in order.

Comments:

The authors took into account a number of my comments. In general, it could be a “Minor revision”. However, I consider the issue related to the classification of bacteriophages to be important.

-The authors provide a classification according to the Master species list 37. However, the Master species list 38 was released quite a long time ago. The classification of bacteriophages in MSL 37 and MSL 38 has considerable differences. It would be much more useful for the reader to have up-to-date information. In addition, the authors write in MS (lines 69-71) that in the classification of bacterial viruses according to the MSL 37 there are only four orders. This is not true since these four orders only relate to the class Caudoviricetes. Other taxonomic groups of bacterial viruses also have orders. The authors should very carefully revise the MS text regarding taxonomy and provide up-to-date data.

Answer: This has been corrected by amending the information in lines 68-73. The paper discussing the new classification has been referenced (reference number 18).  

Minor  points:

-line 73. There is no need to underline the family name.

Answer: Corrected (line 75)

-line 172. the word “Siphovirus” should be written with a small letter in a normal font.

Answer: Corrected (line 173).